# High-Resolution Single-Pixel Imaging of Spatially Sparse Objects: Real-Time Imaging in the Near-Infrared and Visible Wavelength Ranges Enhanced with Iterative Processing or Deep Learning

**DOI:** 10.3390/s24248139

**Published:** 2024-12-20

**Authors:** Rafał Stojek, Anna Pastuszczak, Piotr Wróbel, Magdalena Cwojdzińska, Kacper Sobczak, Rafał Kotyński

**Affiliations:** 1Faculty of Physics, University of Warsaw, Pasteura 5, 02-093 Warsaw, Poland; rafal.stojek@fuw.edu.pl (R.S.); anna.pastuszczak@fuw.edu.pl (A.P.); piotr.wrobel@fuw.edu.pl (P.W.);; 2VIGO Photonics, Poznańska 129/133, 05-850 Ożarów Mazowiecki, Poland

**Keywords:** single-pixel imaging, infrared imaging, compressive imaging, computational imaging, image reconstruction algorithms, deep learning, signal processing

## Abstract

We demonstrate high-resolution single-pixel imaging (SPI) in the visible and near-infrared wavelength ranges using an SPI framework that incorporates a novel, dedicated sampling scheme and a reconstruction algorithm optimized for the rapid imaging of highly sparse scenes at the native digital micromirror device (DMD) resolution of 1024 × 768. The reconstruction algorithm consists of two stages. In the first stage, the vector of SPI measurements is multiplied by the generalized inverse of the measurement matrix. In the second stage, we compare two reconstruction approaches: one based on an iterative algorithm and the other on a trained neural network. The neural network outperforms the iterative method when the object resembles the training set, though it lacks the generality of the iterative approach. For images captured at a compression of 0.41 percent, corresponding to a measurement rate of 6.8 Hz with a DMD operating at 22 kHz, the typical reconstruction time on a desktop with a medium-performance GPU is comparable to the image acquisition rate. This allows the proposed SPI method to support high-resolution dynamic SPI in a variety of applications, using a standard SPI architecture with a DMD modulator operating at its native resolution and bandwidth, and enabling the real-time processing of the measured data with no additional delay on a standard desktop PC.

## 1. Introduction

Single-pixel imaging (SPI) [1,2] has led to a multiplicity of novel ideas about image measurement at various wavelength ranges, spectral imaging, imaging through scattering media, 3D imaging, etc. [3,4,5]. SPI is designed to capture images using a single detector, instead of the millions of detectors (pixels) included in traditional cameras. This enables imaging in wavelength ranges where conventional multi-pixel sensor arrays are expensive or even unavailable (e.g., infrared [6,7,8,9,10], ultraviolet [11], terahertz [12,13], or X-ray [14]). Additionally, SPI makes it possible to capture images through scattering or opaque media, including fog or biological objects, where conventional imaging techniques often fail due to light scattering.

High-resolution SPI (in the meaning of the number of pixels) puts immense demands on the image modulation bandwidth, and the popular DMD modulators operating at frequencies on the order of 20 kHz are often used at actual resolutions between 32 × 32 and 256 × 256, which is a fraction of their capability. In fact, with no compression and in the most straightforward differential imaging mode with a doubled number of expositions, SPI imaging at the resolution of 32 × 32 pixels is possible at only 10 Hz (since displaying 32×32×2=2048 patterns at 20 kHz takes 0.1 s). The acquisition of a single high-quality 1024 × 768 image takes a lot more time. For instance, in a recently proposed fast Fourier SPI method [15], it required as much as 1180 s, which is close to four orders of magnitude more than in our present paper. Long acquisition times may be hard to accept in practical applications. However, compressed SPI imaging has been shown many times at resolutions up to 256 × 256 and at frequencies between 5 and 15 Hz [10,16,17,18,19,20]. Unfortunately, with high compression, image reconstruction becomes computationally demanding. In this paper, we push SPI imaging with DMD modulator to its limits, going for the native 1024×768 binary mode resolution, with efficiently implemented differential sampling and image reconstruction. The cost of this approach is accepting extreme compression, and only sparse images can be reconstructed accurately.

In recent years, there has been growing interest in applying deep learning methods to compressive SPI [21]. In [17], an end-to-end solution is proposed, where a convolutional autoencoder generates sensing patterns and reconstructs images from sparse SPI measurements in real time. Recent studies have explored the use of neural networks (NNs) in SPI systems, either for end-to-end image reconstruction [22,23,24] or as denoising tools that complement other reconstruction methods [25,26]. The latter approach offers more flexibility, as the NN can be easily adjusted to work with various sampling schemes, compression levels, and changing measurement conditions. Additionally, there is growing interest in combining convolutional and recurrent NN to reconstruct temporal sequences of images from dynamic compressive measurements [27,28]. Although NNs provide flexible, fast, and increasingly accurate solutions to inverse problems, they also have notable disadvantages. The training process is typically time-consuming and demands significant GPU power and memory. Moreover, to achieve high-quality reconstructions and sufficient generality, an NN must be trained on large, diverse datasets, which can pose a challenge in some scenarios. To address these issues, there has been interest in exploring the possibility of applying untrained NNs to SPI image reconstruction [29,30,31].

In this paper, we propose a framework for fast, high-resolution single-pixel imaging applicable to spatially sparse objects. This work extends our previous study [32,33], where we introduced a sampling scheme based on image maps and a two-stage image reconstruction approach. The sampling is now modified to distribute information about the mean value of the object across all sampling patterns. The iterative reconstruction algorithm is reformulated with minor adjustments in matrix form for more straightforward implementation on GPUs. Additionally, we have developed Python code using the PyTorch module, enabling GPU execution. We also introduce an alternative second-stage reconstruction method based on a proposed NN, implemented in PyTorch. This latter approach is significantly faster but lacks the generality of the iterative algorithm. Finally, the proposed algorithms are validated in an optical setup with two parallel channels, with detectors operating in the VIS and NIR wavelength ranges, respectively.

The paper is organized as follows. In the Materials and Methods section, we provide separate subsections to explain the consecutive elements of the SPI framework. In Section 2.1, Section 2.2 and Section 2.3, we describe the development of sampling patterns and the motivation behind this approach. In Section 2.4, we explain the initial stage of the reconstruction algorithm, which relies on the generalized inverse of the measurement matrix. Section 2.5 and Section 2.6 detail the two alternative approaches for the second reconstruction stage, and Section 2.7 describes the optical setup. In the Results section, in Section 3.1, we outline the computational and bandwidth requirements and limitations. In Section 3.2, we discuss various noise sources, and we demonstrate cases where the proposed SPI framework produces high-resolution results. Finally, in Section 3.3, we present our experimental results. The paper concludes with the discussion in Section 4.

## 2. Materials and Methods

Sampling patterns and the image reconstruction algorithm are the two components that define any SPI framework. They are often independent, and various image reconstruction algorithms may be used in conjunction with different sampling patterns [4,12,34,35]. The block diagram of the SPI framework proposed in this paper is shown in Figure 1, and the corresponding SPI optical setup is shown in Figure 2. The details of the SPI operation are presented in the following subsections.

### 2.1. Constraints and Objectives for the Proposed Sampling Framework: SPI Pattern Requirements for DMD Modulators

The type of image sampling is tied to the hardware used for spatial modulation. If DMD technology is used for this purpose, the sampling patterns should be binary, as the DMD mirrors inherently have only two states. Using grayscale modulation requires introducing spatial or temporal multiplexing, which occurs at the cost of the native DMD resolution or bandwidth [36]. The DMD bandwidth is also harnessed by utilizing differential projection [36], a commonly used technique that improves the SNR, removes background light from the detector signal, and enables the display of both the positive and negative patterns present in, for example, Walsh–Hadamard matrices. However, these objectives can largely be achieved by adding a single pattern to those displayed, rather than doubling their number [37].

There are other important challenges introduced in a realistic optical setup that are not present in a purely digital experiment. These include the need for noise-mitigating techniques such as differential or complementary detection [38,39,40], as well as maintaining a consistent detection signal range throughout the measurement [10]. The latter suggests that it is inadvisable to combine, in a single measurement, sampling patterns where all pixels are in a single state with those where approximately half of the pixels share the same state. Otherwise, the experiment results in the suboptimal usage of the detector’s dynamic range and A/D converter’s bit depth for signal acquisition. Additionally, a signal obtained with a small number of pixels in the on state tends to be weak in low-light environments and does not take advantage of Fellgett’s [41,42] multiplex advantage, which is particularly important in the infrared wavelength range.

To address these hardware-driven requirements, the sampling patterns need to satisfy the following properties:Sampling patterns should be binary (with values 0 and 1) because DMD mirrors have two states.Sampling patterns should include approximately half of the pixels in the on state (due to the arguments related to light efficiency, Fellgett’s advantage, level of quantization errors induced by DAQ, and signal entropy considerations).Sampling should be differential, meaning that image reconstruction is based on the differences between measurements with consecutive patterns. This eliminates any constant bias from stray light, the background, or inactive areas of the DMD.Sampling patterns should have a spatial spectrum dominated by low spatial frequency contents because most real-world images have their Fourier representation concentrated at low spatial frequencies (this is the usual approach in compressive SPI with, for example, the selection of a subset of Walsh–Hadamard, DCT or wavelet patterns).Sampling should provide easily accessible information on the locations of sparse areas of the probed image. Identifying sparse image regions makes it possible to reconstruct dense regions with better accuracy also at a low sampling rate. For instance, random binary patterns would measure mostly the mean value of the entire image. The best would be patterns in the forms of small figures with various shapes.

The last postulate contradicts some of the previous ones. For instance, any small figure certainly includes less than half of the image pixels. This is why we propose a sampling scheme that involves differential multiplexing and is based on image maps. The measurement with these patterns provides data equivalent to a different measurement that would include patterns with small binary figures.

### 2.2. Sampling Patterns

The process of preparing the sampling patterns and storing them in the measurement matrix (M) is described in Algorithm 1. The entry point to the algorithm is a set of arbitrary image maps. Every image map is a mapping between the image pixels and a subset of integer numbers, thereby defining a partition of the image plane into subareas or regions that have been assigned equal numbers. A single region may include more than one group of adjacent pixels or even isolated pixels. These partitions are introduced in the form of image maps. Here, we use uniformly distributed maps with varying characteristic sizes. The Python code used to generate the maps is the same as the one included in the supplementary materials of Ref. [32]. The maps could be defined in other ways as well; for instance, by analogy to foveated SPI imaging [19,43,44], some regions of interest could be enhanced at the expense of other areas [33]. The typical feature size of the regions can also be aligned with specific object feature sizes and shapes. Sample maps are shown in Figure 3. We usually take l=100 maps, each with m=31 regions. Then, the algorithm encodes a sequence of k=l·(m+1)=3200 binary full-resolution 1024×768 patterns. This corresponds to the sampling rate (compression rate, i.e., the ratio of the number of sampling patterns to the number of pixels) of 0.41.

Samples of encoded patterns are shown in Figure 4. Encoding or translating image maps into patterns involves using a lookup table, which is described in the next subsection. The lookup table used throughout this paper is shown graphically in Figure 5.
**Algorithm 1** Construction of a binary measurement matrix from image maps.**Input:** m—l×n matrix containing *l* maps with $n=n_{x}\cdot n_{y}$ pixels**Input:** r—vector of length *n* with random integer values in the range [1,l]**Input:** A—a look-up table; (m+1)×m binary matrix, such that D·A is full rank. Here D is finite difference operator that subtracts matrix rows.**Output:** M—k×n measurement matrix with rows containing binary patterns with *n* pixels each **function** make patterns(m,r,A)     M←initializeandfillwithzeros     p←0     **for** i=1,l **do**▹ Iterate maps         $m_{i}\leftarrow i-\mathrm{th}\;\mathrm{row}\;\mathrm{of}\;m$▹*i*-th map         **for** q=1,(m+1) **do**▹ Iterate rows of A            **for** j=1,m **do**▹ Iterate sectors of map $m_{i}$                p←p+1▹ Index of the next sampling pattern                $v\leftarrow where(\left(m_{i}=j\right)and\left(r\neq i\right))$
▹ Get indices of all pixels of *j*-th sector of *i*-th map                M(p,v)←A(q,j)▹ Assign binary values to pixels of a sector            **end for**         **end for**     **end for**     **return** $\tilde{M}$▹ Return the measurement matrix **end function**

Algorithm 1 is similar, but not identical, to the one we proposed previously [32,33]. There is a small change that affects the way the mean value of the object is encoded in the sampling patterns. Previously, we dropped some rows from the lookup table when translating the second and subsequent maps into patterns. This approach removes redundancy in the sampling. However, the disadvantage is that it allows the recovery of the mean value of the object from the first m+1 sampling patterns. Since the mean value of the object influences all pixels in the reconstructed image, this type of sampling resulted in a very non-uniform distribution of values in the generalized inverse matrix.

Currently, Algorithm 1 processes all the maps in the same way, distributing the information on the mean value of the object across all sampling patterns. As a result, the inverse matrix has improved uniformity.

### 2.3. Differential Multiplexing

The mechanism of differential multiplexing is governed by the multiplexed pattern encoding algorithm, with a crucial role played by the dedicated binary lookup table. The translation of image maps into sampling patterns is described by Algorithm 1. Pattern encoding is performed using a lookup table, which is a binary (m+1)×m matrix A. The table is scanned row by row, with its columns indicating which map regions should be included in the encoded sampling pattern. The encoding mechanism should ensure that the differences between consecutively created binary patterns convey the same information on the sampled objects as would be obtained through sampling using individual map regions. Therefore, the lookup table must satisfy certain mathematical properties. Specifically, its rows should contain approximately equal numbers of zeros and ones, and a matrix formed by subtracting its rows should be a full-rank square matrix. These conditions cannot be achieved for even *m*, so the matrix has an odd number of columns. A gallery of lookup tables of different sizes is shown in Figure A1. Despite these conditions, there is still room for further optimization, and we search for matrices with the smallest possible determinant det(D·A). As a result, the lookup tables we use are derived from brute-force numerical optimization. It is worth noting that the columns of the lookup tables can be freely reordered, but not the rows. In fact, the regions of each map can be reordered as desired.

Finally, we note that in a non-differential measurement, we could use a square Hadamard matrix instead of our lookup tables, as Hadamard matrices are inherently full-rank and have an equal number of the two binary values (with the first row different). In fact, Hadamard matrices have a well-established position in multiplexed measurements. However, Hadamard matrices include both positive and negative values (e.g., ±1), which introduces additional challenges when displaying them on a DMD. To address this, one would need to double the number of exposures, thus reducing the effective DMD bandwidth. Therefore, the proposed matrices are more suitable in designing an inherently differential multiplex measurement that is equivalent to binary (0,1) sampling with patterns occupying small parts of the image (see Figure 6) and together forming *l* independent partitions of the image surface. In many cases, they may be better suited for conducting a multiplexed measurement than Hadamard matrices.

### 2.4. Initial Image Reconstruction

The measurement of an object in SPI imaging is usually expressed mathematically using a concise measurement equation, which in a noise-free situation may be written as
(1)y=M·x.

Here, x denotes the object, such as a 2D image, with pixels gathered into a single vector. M is the measurement matrix whose rows contain the sampling patterns. Finally, y is the measurement vector. A refined version of this equation would also include signal-dependent and signal-independent noise. There are various ways to conduct a differential measurement. In the present work, we calculate the difference of subsequent elements in the measurement vector, i.e., $y^{\prime }=D\cdot y$, and use only $y^{\prime }$ for image reconstruction. Here, D denotes a finite difference operator that subtracts the subsequent rows of a matrix or the subsequent elements of a vector. Any constant DC component, such as one originating from background light, is eliminated from $y^{\prime }$. The SPI reconstruction algorithm should then be able to recover x from D·y. In a compressive measurement, where M is a rectangular matrix with more columns than rows, this recovery is usually ambiguous and approximate.

Most SPI image reconstruction algorithms are either iterative or based on applying the pseudoinverse of the measurement matrix to the measured signal, i.e., $\tilde{x}=M^{+}\cdot y$. This is particularly straightforward when the measurement matrix is semi-orthogonal. An even simpler situation arises when it corresponds to a fast linear transform such as DCT, Fourier, or Walsh–Hadamard transform. Our approach is slightly different. We work with the full form of the measurement matrix M and find its generalized inverse $M^{g}$. The Moore–Penrose pseudoinverse is an example of a matrix’s generalized inverse, but it is not the only one. The calculation of the generalized inverse of M, which is a large matrix, is computationally demanding. It becomes even more difficult as the sampling rate is increased.

$M^{g}$ is calculated before SPI imaging, and it may require a significant amount of memory (we typically work with 10 GB inverse matrices). However, the image reconstruction is fast because it is based on a simple matrix–vector multiplication. Since we assume a differential measurement, we need an inverse matrix of the form $P={\left(D\cdot M\right)}^{g}$.

For the generalized inversion (^g^), one could use the pseudoinverse, but it is preferable to use a regularized pseudoinverse. By choosing various generalized inverses of the measurement matrix, we could calculate different solutions of the underdetermined measurement equation. The role of regularization is to select such a solution that additionally minimizes a combined quadratic criterion. In our case, the criterion is defined as a weighted sum of squared norms of the convolutions with selected spatial filters, including discrete gradient filters, a penalty filter for high spatial frequencies, and an identity filter [16]. We use the name Fourier-Domain Regularized Inversion (FDRI) for this kind of matrix inversion because the quadratic criteria are conveniently written in the Fourier space as positive definite diagonal quadratic forms. With FDRI, it is possible to reconstruct images from a compressive SPI measurement with fewer artifacts than with a simple pseudoinverse. In Ref. [16], FDRI was applied for real-time SPI imaging. The quality of the reconstructed images was comparable to that obtained by the minimization of the total-variation (TV) norm, while the image reconstruction time was almost two orders of magnitude smaller. The form of the generalized inverse used here is the same as in Ref. [16] with the only difference being that we calculate the generalized inverse of D·M [10,32] and not just of M:(2)$$P=F^{*}\cdot \hat{\Gamma }\cdot F\cdot {\left(D\cdot M\cdot F^{*}\cdot \hat{\Gamma }\cdot F\right)}^{+}\cdot D.$$

Here, M is the measurement matrix, F is the 2D Fourier transform, D is the finite difference operator, and $\hat{\Gamma }$ is a diagonal non-negative regularizing term [16,32]. Matrix $\hat{\Gamma }$ is defined as in Ref. [16],
(3)$${\hat{\Gamma }}_{i,j}=\frac{\delta_{i,j}}{\sqrt{{\left(1-\mu \right)}^{2}\left(sin^{2}\left(\omega_{x}\right)+sin^{2}\left(\omega_{y}\right)\right)+\mu^{2}\frac{\omega_{x}^{2}+\omega_{y}^{2}}{2\pi^{2}}+\epsilon }},$$
where μ and ϵ are used to weight the criteria included in the regularization, and $\omega_{x,y}$ are the spatial frequencies. We assume that μ=0.5 and $\epsilon =10^{-7}$.

The reconstructed image is then calculated as
(4)$$\tilde{x}=ReLu\left(\tilde{x_{0}}\right),$$
where
(5)$$\tilde{x_{0}}=P\cdot y.$$

The ReLu function removes the negative part from the reconstructed image, which we consider to be a reconstruction artifact. At lower resolutions, $\tilde{x}$ could be the final result of SPI imaging. Since we are interested in high-resolution SPI, this first-stage reconstruction serves as the starting point for further enhancement with Algorithm 2 or an NN.
**Algorithm 2** Image reconstruction from a compressive measurement.**Input:** y—measurement vector of length *k* (it i assumed that k≪n, where $n=n_{x}\cdot n_{y}$ is the image size in pixels, and that the measurement equation is y=M·x, where x is the measured image, and M is the binary measurement matrix)**Input:** V—l×m×n binary array such that $V_{i,j,p}=1$ at pixels *p* belonging to region *j* of map *i* unless $r_{p}=i$. In further notation, $V_{i}$ (i∈{1,2,…,l}) represents a m×n slice of V which corresponds to the *i*-th map.**Input:** W—array of *l* matrices $W_{i}=V_{i}\cdot P$ (i∈{1,2,…,l}) with dimensions m×k. Here $P={\left(D\cdot M\right)}^{g}$ is the generalized matrix inverse ^g^ applied to the measurement matrix M after taking the differences of its rows with operator D**Input:** $\tilde{x}$—the initial image reconstruction vector of size $n=n_{x}\cdot n_{y}$ (set by us to the initial reconstruction result ReLu(P·y) but may also be filled with constant positive values)**Input:** f∈(0,1)—learning rate (we took f=0.75)**Output:** $\tilde{x}$—vector of size *n* with the reconstructed image **function** IMAGE RECONSTRUCT(y, W, V, $\tilde{x}$, *f* )     **for** i=1,maxiter **do**         **for** j=1,l **do**▹ Loop over maps            $w_{j}\leftarrow W_{j}\cdot y$▹ Expected pixel sums in sectors of map            $v_{j}\leftarrow \;V_{j}\cdot \tilde{x}$▹ Current pixel sums in sectors of map *j*            $c\leftarrow \;w_{j}/v_{j}$ where $v_{j}\neq 0$ else 0            $\tilde{x}\leftarrow \left(1-f\right)\tilde{x}+f\cdot diag\left(c\cdot V_{j}\right)\cdot \tilde{x}$         **end for**        **end for**     **return** $\tilde{x}$▹ Return the reconstructed image **end function**

### 2.5. Reconstruction Enhancement with an Iterative Algorithm

The proposed Algorithm 2 enhances the initially reconstructed image by leveraging two key elements. First, since the initial reconstruction is based on the generalized inverse of the measurement matrix, P·y exactly fulfills the measurement Equation (Equation 1). Second, the composition of the binary measurement matrix M ensures that differential sampling with patterns from this matrix is mathematically equivalent to using patterns that correspond to all individual regions of all image maps. In other words, in the absence of measurement noises, the reconstructed image is guaranteed to have the correct mean values within every region of every map. Empty regions will have the mean value of zero. Because the images have only non-negative pixel values, a mean value of zero (or near zero) signifies that the region is empty.

Algorithm 2 iteratively processes all image maps, applying corrections to the mean values of pixels within the regions of each map (this is the “Loop over maps” in Algorithm 2). Because these regions are non-overlapping, they can be processed in parallel. A similar reconstruction method was used in Ref. [32]; however, here we introduce a parameter, the learning rate 0<f<1, which typically improves convergence in the presence of noise. In this paper, we assume f=0.75.

An important property of this algorithm is its efficient elimination of empty (zero-valued) regions from the reconstructed image. As a result, for spatially sparse images, it improves the image quality in regions containing objects. This improvement is achieved because the limited information in the compressed measurement y is directed toward reconstructing only the non-empty regions of the image. This approach can be viewed as complementary to adaptive sampling; however, adaptive sampling is technically more complex, requiring additional logic in the measurement stage and posing challenges in generating adaptively high-resolution patterns at the DMD frame rate. In contrast, our approach relies on predefined patterns for measurement and separates the measurement process from the reconstruction phase.

### 2.6. Reconstruction Enhancement with a Neural Network

In an alternative approach to Algorithm 2, we explore the application of an NN to enhance the reconstruction quality (see Figure 7). The proposed network bears some resemblance to the U-Net architecture; however, instead of concatenation in the skip connections, we employ weighted data summation, where the weights are learned parameters of the network. This modification enables the network to balance the influence of input data and extracted features on the final output, placing it in a functional space between a U-Net and an autoencoder.

The NN is trained simultaneously on two handcrafted image datasets. The first dataset comprises binary, high-resolution images of line art, while the second dataset contains images of randomly placed and scaled handwritten digits drawn from the MNIST dataset [45]. The training dataset consists of 30,000 images, drawn equally from both datasets, and is randomly regenerated in each training epoch to enhance the variety of the training data. The evaluation dataset, which is precalculated, contains 6000 images, also drawn equally from both datasets. Notably, the handwritten digits included in the evaluation dataset are not used for generating the training images. All images in both datasets have a resolution of 768×1024 pixels and are normalized to pixel values in the range [0, 1]. The generated images contain 96% of zero-valued pixels on average (ranging between 89% and 99.7%).

The NN is designed to perform the second-stage reconstruction as an alternative to using Algorithm 2. The input data consist of initially reconstructed images obtained by applying Equation (Equation 5) to simulated SPI measurements of the training or testing images. To model realistic experimental conditions, we introduce additive Gaussian noise to the measurement. The network is trained to optimize a joint loss function:(6)$$C=C_{1}+\lambda \cdot C_{2},\;\;\;\;\mathrm{where}\;C_{1}=MSE\left(\hat{x},x\right),\;\;C_{2}=MSE\left(\hat{v},v\right),$$
where MSE denotes the mean squared error, *x* represents the pixel values of the ground truth image, $\hat{x}$ is the output of the NN, *v* and $\hat{v}$ are the sums of pixel values within all sectors of the maps calculated for images *x* and $\hat{x}$, respectively, and λ is a weight parameter empirically adjusted during training.

### 2.7. Optical Set-Up

We validate the performance of the proposed algorithm by implementing the SPI optical setup using a 1024×768 pixel DMD (Vialux V-7001 XGA with DLP7000), which splits the light beam into two separate detectors: an uncooled Thorlabs PDA100A2 silicon photodiode for the 300–1100 nm range, and a thermoelectrically cooled VIGO Photonics PVI-4TE-3 detector for the 1100–3000 nm range. The schematic is shown in Figure 2. Light from a thermal source passes through a test pattern composed of a series of holes in a nontransparent metal foil, with the object observed in transmission mode.

To minimize chromatic aberration, we first use a set of gold-coated parabolic mirrors as the imaging system. We also test an alternative setup using fused silica lenses. The parabolic mirrors provide better signal-to-noise ratio (SNR) for the infrared (IR) optical channel, while the glass lenses offer higher image resolution in the visible light range. The signal is digitized using a 14-bit digital oscilloscope (Picoscope 5000) with a sampling at 128 ns. Since the DMD frequency is lower, the measurement vector y is obtained from these samples by averaging.

## 3. Results

### 3.1. Image Acquisition and Reconstruction Times and Computational Requirements

Throughout this paper, we assume an image resolution of 1024×768, with the object being sampled at a compression rate of 0.41%, requiring 3200 binary patterns for sampling. The DMD operates at 22 kHz, resulting in an image acquisition frequency of 6.9 Hz. Our image reconstruction algorithm is implemented in Python, utilizing the NumPy and PyTorch libraries, and is capable of running on both CPU and GPU processors. Due to the high memory demands, we use a GPU with 24 GB of memory (NVIDIA GeForce RTX 3090). Specifically, the measurement matrix M and the inverse matrix P each occupy 10 GB in 32-bit floating-point format. This suggests that further increasing the number of sampling patterns at the same resolution may present challenges for this framework and could require either a scanning strategy or parallel SPI measurements in lower-resolution blocks.

The initial reconstruction takes 16 ms and can be performed on the fly without a noticeable impact on the DMD acquisition rate, reducing the rate from 6.9 Hz to 6.2 Hz. The second-stage reconstruction using the NN takes 26 ms (including the initial reconstruction stage), resulting in an SPI rate of 5.8 Hz. The iterative algorithm is slower, requiring approximately 50 ms per iteration, which limits real-time applications to one or two iterations; performing 50 iterations would take 2.5 s. Additionally, due to the GPU architecture, images from two or more channels can be reconstructed in parallel without additional time penalty, leveraging the batch processing implemented in PyTorch. Without a GPU, the processing times are significantly longer. On an Intel i7-11700K CPU, the initial reconstruction takes 350 ms, and the iterative algorithm is much slower, taking 2.2 s per iteration (however, the computation speed could improve by up to tenfold with a Numba-based implementation [32]). In summary, for applications that are not time critical, Algorithm 2 may be applied with multiple iterations. At the same time, only a GPU-based approach—either using the second-stage reconstruction with the NN or a single iteration of Algorithm 2—enables image acquisition and reconstruction at over 6 Hz at the resolution of 1024×768 without hindering the DMD acquisition rate.

### 3.2. Effect of Compression, Background Noise, and Detector Noises on Imaging

Eliminating additive background noise is critical in optical SPI. This noise may include a constant or slowly varying bias in the detector signal due to the dark current, background illumination, or reflections from inactive parts of the DMD, as well as variations in the light source intensity. In infrared imaging, the changing temperature of the setup, and particularly of the DMD, also contributes to a slowly varying additive background signal. Background noise is usually reduced through differential or interferometric techniques, such as projecting negated DMD patterns or using the complementary detection of the two beams reflected by the DMD in the directions corresponding to the on and off states of the mirrors. In our case, we incorporate the differential mechanism in the way we construct sampling patterns and measurements. Thus, the reconstructed image is invariant with respect to adding any constant value to the detection signal. This approach also allows us to use the AC coupling of the DAQ, although this type of high-pass filtering is not entirely transparent to the image reconstruction outcome.

A variety of other noise sources also affect SPI imaging, and the impact of one source may often become dominant. We now discuss the effect of compression and detector additive noise. The impact of compression on SPI imaging quality depends on the structure of the object. In our work, the compression rate is extremely small, and the reconstruction quality depends on the presence of empty (zero-valued) areas within the object image. To our knowledge, there are no existing test image datasets categorized by spatial sparsity, making it difficult or impossible to use standard image archives for the analysis in this paper. Consequently, we first select three images with different properties to illustrate the typical behavior of the presented SPI method in the presence of different sparsity levels. We then create synthetic images based on the MNIST database with added Gaussian noise for further comparisons.

In Figure 8 and Figure 9, we demonstrate the role of spatial sparsity in SPI imaging with three example images: the Siemens star, which is a dense grayscale test object without empty regions (with less than 0.1% zero-valued pixels); a limited-field-of-view Siemens star (with 96.3% of zero-valued pixels); and a sparse line-art image (with 98.9% of zero-valued pixels). In this analysis, due to the absence of other noises, compression is the only factor responsible for imperfect imaging. An approximate reconstruction result is obtained with the initial reconstruction method, similar to what could be achieved with low-resolution SPI imaging (e.g., 128 × 128 or 256 × 256 at most). For the grayscale image, no significant improvement is achieved with the iterative algorithm. However, Algorithm 2 significantly improves the quality of the other two images, achieving up to the DMD resolution (1024 × 768). In Figure 8, we also measure the reconstruction quality using two standard criteria: peak signal-to-noise ratio (PSNR) and structural similarity (SSIM). The notable differences in these criteria reveal that averaging these measures across a range of standard database images would obscure information about the properties of imaging sparse objects.

In Figure 9, we show the convergence of PSNR and SSIM for the same three objects as a function of the number of iterations in Algorithm 2. In addition to the data presented with solid lines, we show the same convergence when the initial reconstruction stage is omitted, which slightly improves the computation time and significantly reduces the memory requirements due to the large size of the inverse matrix P. Results from this plot indicate that typically the first 1–5 iterations are sufficient to nearly reach the final image quality. The importance of the first stage reconstruction largely depends on the type of image; it is especially crucial for the grayscale Siemens star, although it also plays a role in achieving convergence of PSNR and SSIM for the other two objects.

The role of additive detector noise is depicted in Figure 10, Figure 11 and Figure 12. An often overlooked source of additive noise is the limited bit resolution of the detection system introduced by the DAQ. Depending on the sampling patterns, the required dynamic range of the detector may be utilized more or less efficiently. A common issue in SPI is that, aside from patterns containing approximately half of the pixels in either the on or off states, there also appear frames with all pixels switched on or off. As a result, the quality of the measurement depends on fine signal changes between consecutive projected patterns, yet some signal points deviate significantly from the typical values, which imposes unnecessary constraints on the DAQ bit resolution. We avoid this issue by properly constructing the measurement matrix. Nevertheless, the DAQ bit resolution does influence the final SPI quality as shown in Figure 10, which illustrates the relationship between PSNR and SSIM versus the DAQ bit resolution.

We note that due to its randomness, DAQ quantization error exhibits properties similar to those of additive Gaussian noise, unless very few bits are effectively used in the signal’s quantization. According to the results in Figure 10, at resolutions below 12 bits, the limited bit resolution begins to impact the image quality more than compression. For this reason, in the optical experiments, we use at least 14-bit quantization in the measurement.

A comparison of the performance of the two second-stage reconstruction methods (Algorithm 2 and NN) in the presence of additive Gaussian detector noise is shown in Figure 11. This comparison is performed on a set of images similar (but not identical) to those included in the NN training set. At an SNR of 45 dB, both methods perform well and achieve high-resolution SPI imaging, with the NN method being significantly faster and yielding better quantitative results in terms of SSIM and PSNR. We use the standard definition of SNR, but since the measurement is differential, we subtract the mean value from the signal before calculating SNR. In fact, adding any constant value to the measurement vector does not affect the reconstructed image.

For images that differ from the training data, the NN often introduces significant artifacts, while Algorithm 2 maintains good performance even for arbitrarily designed images. In Figure 12, we compare these two methods more systematically, using PSNR, SSIM, and the $C_{1}$ and $C_{2}$ loss function components, averaged over a set of 1000 images and plotted against the measurement SNR.

### 3.3. Experimental Results

Real-world visible and infrared scenes often include dense grayscale objects. While the SPI method proposed in this paper can be applied to dense images, high-resolution imaging is only achievable for sparse objects. For dense objects, an alternative measurement matrix—such as one based on binarized DCT patterns—may be more suitable and could improve the overall quality of reconstruction. Sparse objects, in this context, include those obtained through phase contrast or polarization contrast, as well as objects observed over a known background that can be subtracted from the measurement, or scenes with the field of view reduced by vignetting to a selected region of interest.

In our example, a sparse object is represented by a metal foil with small holes arranged in the shape of the faculty logo. In our experimental setup, two measurement vectors are captured simultaneously by two detectors operating in different wavelength ranges. These detectors measure the light intensities reflected from the DMD in two directions corresponding to the on and off states of the micromirrors. Such an arrangement could support complementary detection—a technique for enhancing the signal-to-noise ratio (SNR) and removing the DC component from the signal. Since our measurement is inherently differential, we can utilize the two channels independently. Given that the image reconstruction method is invariant to adding a constant to the signal, data from both channels are processed identically, with the only difference being that signals obtained with mirrors in the negated state (in relation to the measurement matrix, in a boolean sense) also need to be negated (in the sense of multiplication by −1). Grayscale images from the two channels are then used to construct a false-color image.

Figure 13 shows the reconstruction result of the logo image. Figure 14 presents the same image measured during movement. Because SPI is a time-multiplexing technique, non-static objects tend to produce artifacts. Although the resolution is affected, the contents remain clearly visible.

## 4. Discussion

We propose and compare two methods for SPI imaging at the native DMD resolution. Both methods use the same differential multiplexing sampling scheme and an initial reconstruction stage based on the dot product with the regularized generalized inverse of the measurement matrix (according to Equation (Equation 4)). The second reconstruction stage is either based on an iterative Algorithm 2 or an NN, inspired by the convolutional U-Net architecture as shown in Figure 7. The proposed SPI approaches rely on extremely high compression, which is dictated by the limited bandwidth of the DMD modulators. This restricts the possibility of high-resolution reconstruction to spatially sparse images with limited information content. Dense grayscale images, on the other hand, can only be reconstructed with lower accuracy.

For NN-based reconstruction, high-quality results are achieved only for images similar to those in the training set. The NN is trained on line-art images and transformed alphanumeric objects from the MNIST dataset, and may not perform well on other types of objects or images at different resolutions without further training or modifications. In contrast, Algorithm 2 can be directly applied to different resolutions and sparse images with various shapes of the zero-valued areas.

The proposed method is validated in an optical setup with two detectors operating in the VIS and NIR wavelength ranges. The object is sampled at a compression of 0.41%. Both the NN and Algorithm 2, when used with a single iteration, can reconstruct images at a rate that does not hinder the acquisition frame rate. However, Algorithm 2 with multiple iterations can still be applied in practical applications, with a time penalty of 50 ms per iteration on a GPU.

One advantage of Algorithm 2 is its ability to identify empty regions of the object, which helps in improving the reconstruction accuracy in areas containing actual content. In this way, the proposed high-resolution SPI framework offers an alternative to adaptive sampling methods, which are challenging to implement. Given the low resolution of most current SPI systems (typically between 32 × 32 and 256 × 256), which limits their potential to replace traditional high-resolution cameras, we consider this work an important step toward enabling practical SPI applications across a variety of fields. Applications involving the imaging of sparse objects, in particular, may benefit from this work. These include imaging with phase contrast or polarization contrast, as well as cases where objects are observed against a known background that can be subtracted from the measurement. Additionally, it has potential applications in the microscopic imaging of scenes where the field of view is reduced by vignetting to a selected region of interest.

## Figures and Tables

**Figure 1 sensors-24-08139-f001:**
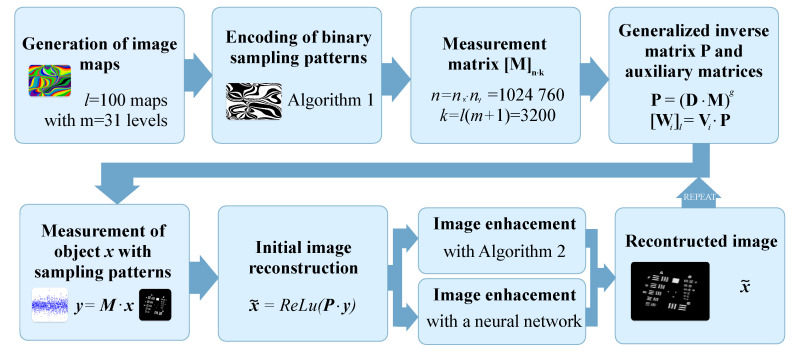
Block diagram of the proposed high-resolution SPI framework. The blocks in the upper row show the preparation stage, involving the calculation of large matrices needed for object sampling and for the reconstruction of the SPI measurement afterward. The blocks in the bottom row describe the actual SPI measurement and reconstruction. In continuous SPI imaging, the bottom blocks are executed cyclically.

**Figure 2 sensors-24-08139-f002:**
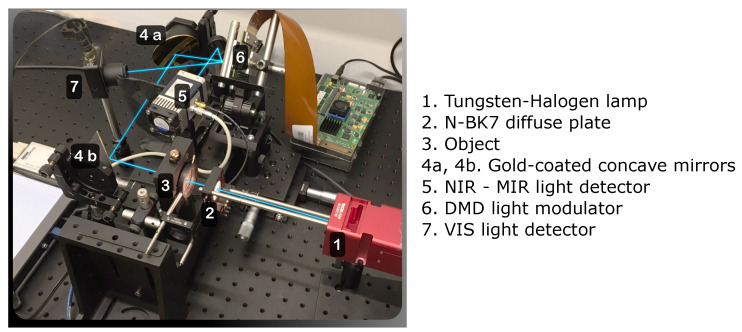
SPI optical setup with a broadband light source, a DMD modulator operating in the native 1024×768 binary mode at up to 22 kHz, and two detectors—a VIS/NIR range Si amplified detector and a thermoelectrically cooled MCT detector for the NIR/MIR range. The detectors are placed in a complementary architecture, measuring the signal reflected by the DMD mirrors in either the “zero” or “one” position.

**Figure 3 sensors-24-08139-f003:**
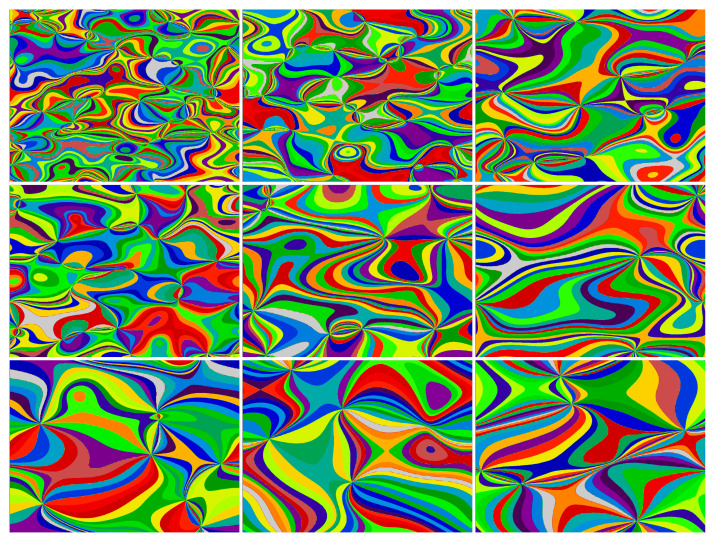
Examples of high-resolution (n=1024×768) image maps. The above 9-element sample is a subset taken from the l=100-element set of maps. Every map includes m=31 different regions shown in different (false) colors. The characteristic feature size is different in each map. Each of the maps is used to create m+1 sampling patterns (see Algorithm 1).

**Figure 4 sensors-24-08139-f004:**
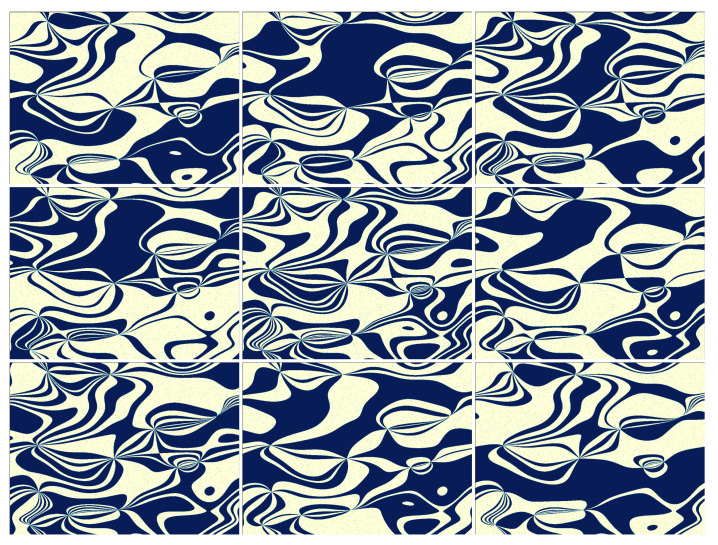
Examples of 1024×768 binary sampling patterns constructed from a single image map (that map is shown in the top-right subplot of Figure 3). The patterns contain approximately the same number of pixels in the on and off states. In total, we calculate l·m=3200 binary sampling patterns, which are then stacked in the measurement matrix M. The sampling patterns are projected onto the DMD in the native binary full-resolution mode. The measurement is differential in the sense that only the differences between subsequent intensity measurements are processed, eliminating the influence of background illumination, yet information about the intensity in each region of every image map is guaranteed to be preserved.

**Figure 5 sensors-24-08139-f005:**
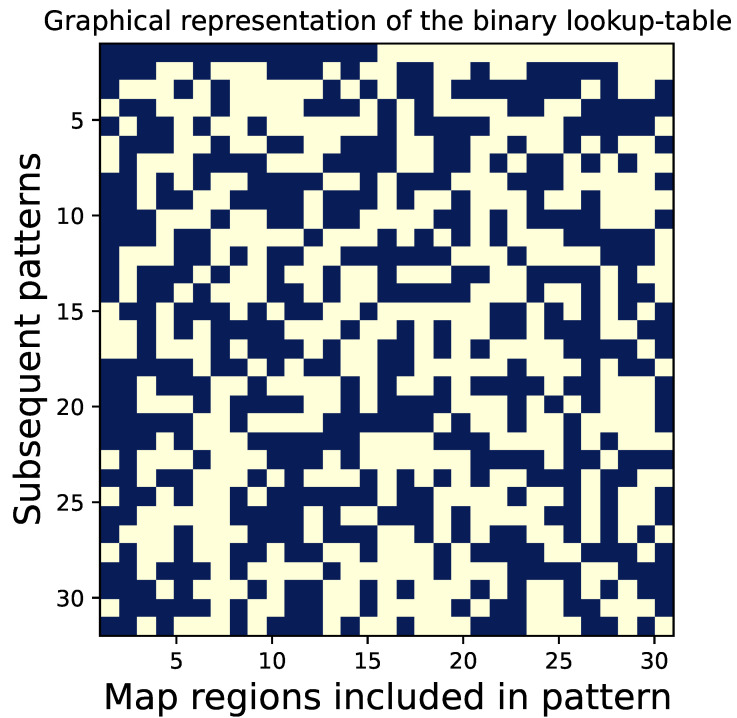
Graphical representation of the binary lookup table A required by Algorithm 1. A is a (m+1)×m matrix with binary values (here, m=31) obtained through numerical optimization [32]. Algorithm 1 uses this matrix to construct m+1 subsequent rows of the measurement matrix M by taking consecutive rows of A and treating its columns as an indication of which regions from the map should be included in the sampling pattern.

**Figure 6 sensors-24-08139-f006:**
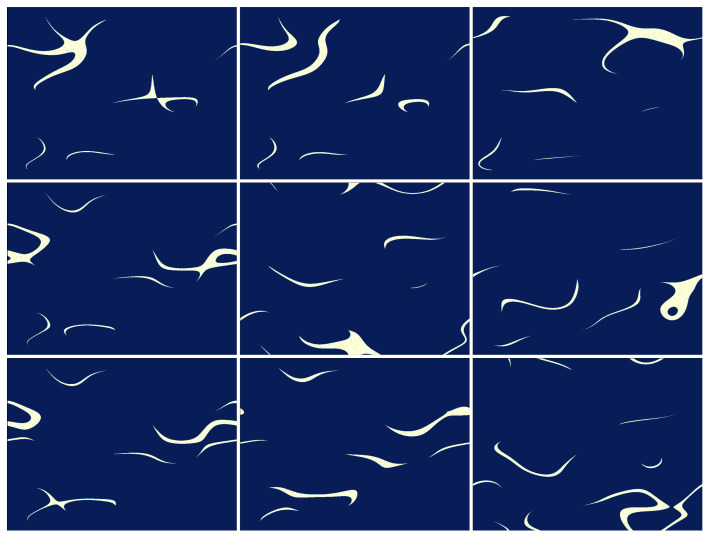
Examples of binary regions extracted from a single image map (that map is shown in the top right subplot of Figure 3, and the sampling patterns shown in Figure 4 include superpositions of the binary, non-overlapping regions of the same map). In a mathematical SPI experiment with no necessity of eliminating a DC background bias and with no limiting effect of the small region size on SNR, the binary regions shown in this figure could be used equivalently to the patterns from Figure 4, and both measurements would acquire exactly the same information on the sampled object. In an optical experiment, the patterns from Figure 4 have the advantage of occupying approximately half of the DMD surface and of the inherent possibility of a differential measurement.

**Figure 7 sensors-24-08139-f007:**
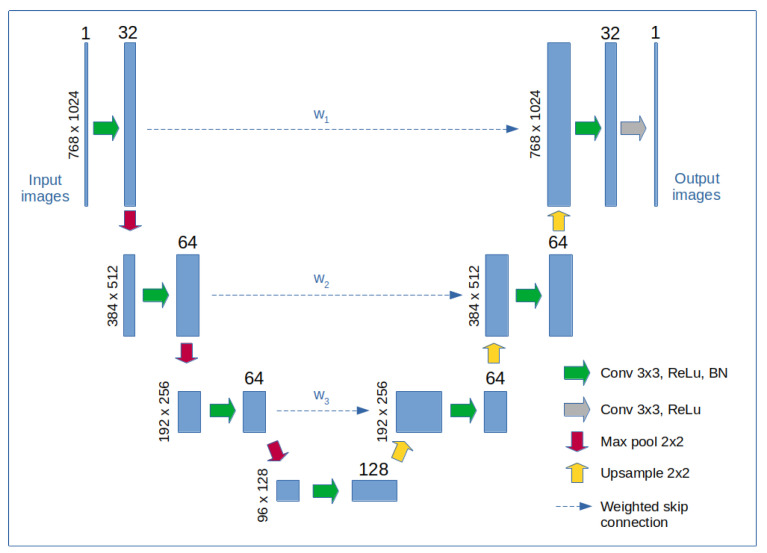
Block diagram of the NN architecture. The network consists of 8 convolutional blocks arranged in a U-Net architecture, with the first 4 blocks forming the contracting path and the remaining 4 blocks forming the expanding path. Each convolutional block comprises a 3×3 convolutional layer, a rectified linear unit (ReLU) nonlinear activation, and a batch normalization (BN) layer. Two-dimensional max pooling is applied in the contracting path to reduce the spatial size of the data, while nearest-neighbor interpolation is used in the expanding path to upsample the data. Additionally, weighted skip connections, with learned weights $w_{1},w_{2},w_{3}$, combine features from matching blocks in both paths to improve data fidelity. The skip connections are applied directly after the convolutional layers and before the ReLU activations.

**Figure 8 sensors-24-08139-f008:**
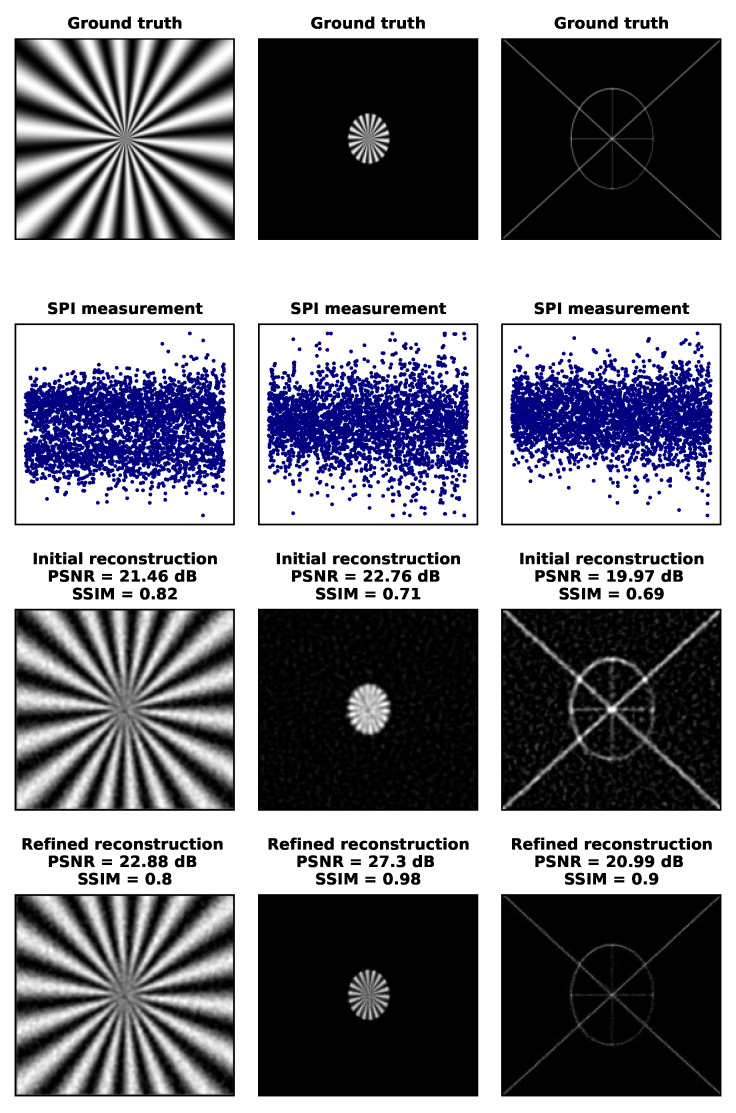
High-resolution (1024×768) SPI imaging at very high compression (k/n=0.41%). The three columns show examples of three different types of objects: **left**—grayscale image (Siemens star) occupying the entire image area, **center**—grayscale image with a limited support region, and **right**—binary, high-resolution, highly sparse line-art image. The rows, starting from the top, show the ground truth, the SPI measurement data, the first-stage reconstruction result, and the second-stage improved reconstruction result (50 iterations). The reconstruction quality of sparse images is significantly improved in the second reconstruction stage with Algorithm 2.

**Figure 9 sensors-24-08139-f009:**
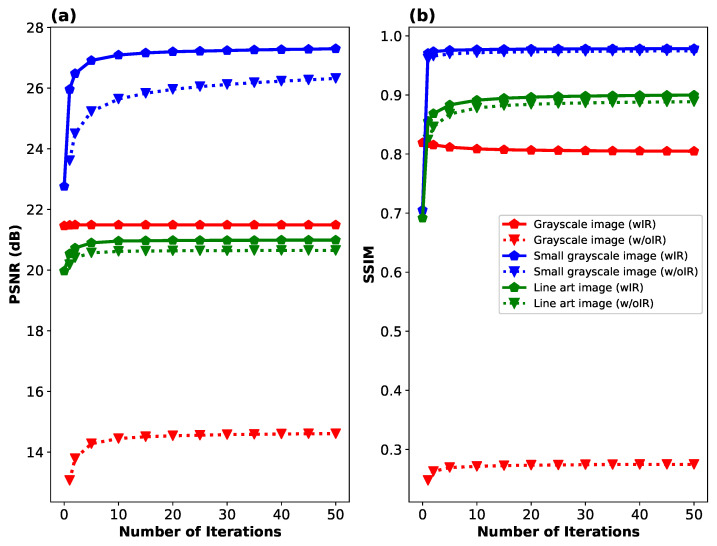
Convergence of image reconstruction quality in terms of two metrics: (**a**) PSNR and (**b**) SSIM as a function of the number of iterations in Algorithm 2. Two variants of the reconstruction algorithm are considered: one with an initial reconstruction calculated using Equation (Equation 4) (wIR), and the other without an initial reconstruction, where the vector $\tilde{x}$ is initialized with constant positive values (w/oIR). The performance is evaluated on three distinct images illustrated in Figure 8: a full-area grayscale image (Siemens star), a smaller grayscale image with limited field of view, and a high-resolution sparse line-art image.

**Figure 10 sensors-24-08139-f010:**
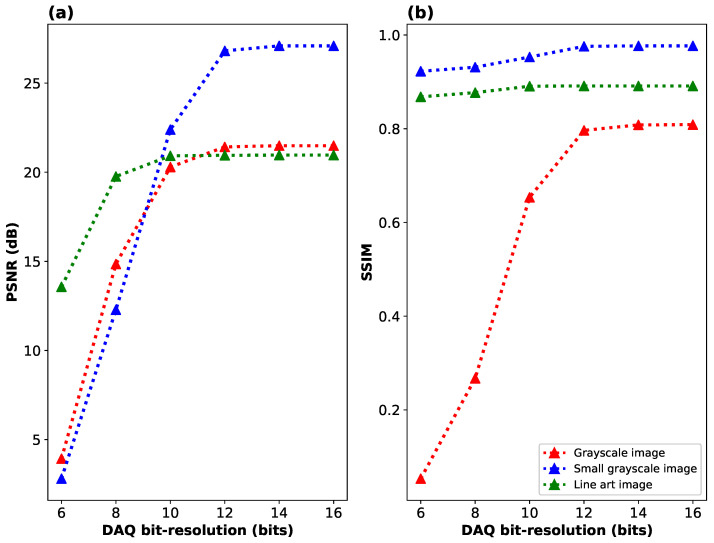
Performance of Algorithm 2 as a function of the bit resolution of the data acquisition device in terms of two metrics: (**a**) PSNR and (**b**) SSIM. The evaluation is performed on three distinct images illustrated in Figure 8: a full-area grayscale image (Siemens star), a smaller grayscale image with a limited field of view, and a high-resolution sparse line-art image.

**Figure 11 sensors-24-08139-f011:**
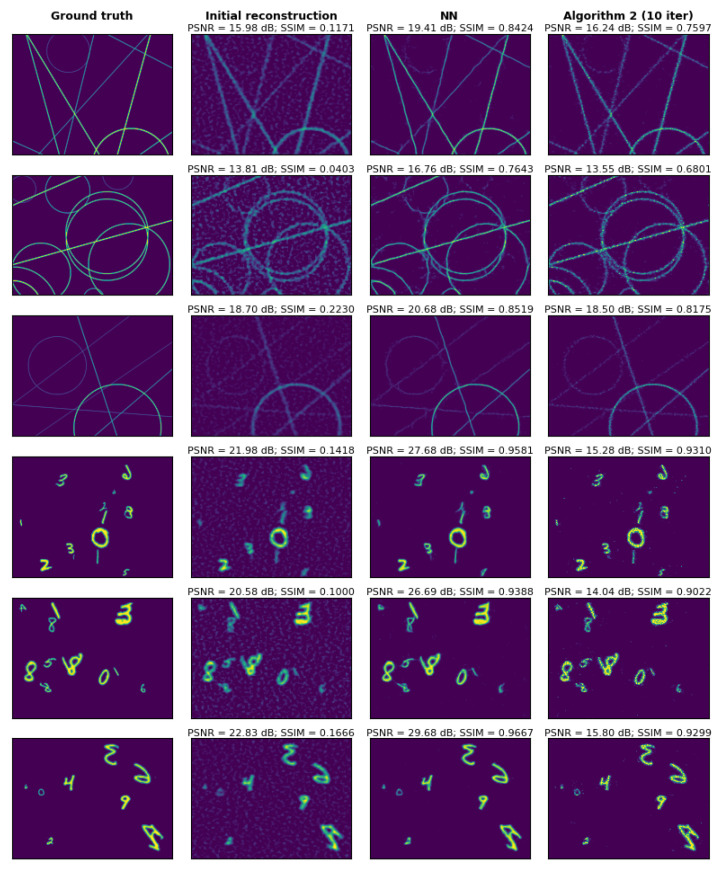
Examples of image reconstructions obtained using either Algorithm 2 or the NN as the second-stage reconstruction method. In this case, a simulated compressive SPI measurement is performed using several exemplary images from the NN evaluation dataset as objects. Additive Gaussian detector noise (with SNR = 45 dB) is included in the measurement. Columns, starting from the left, display the following: (1) the ground truth images, (2) the initial reconstructions obtained using Equation (Equation 5), (3) reconstructions enhanced with the NN, and (4) reconstructions enhanced by applying 10 iterations of Algorithm 2.

**Figure 12 sensors-24-08139-f012:**
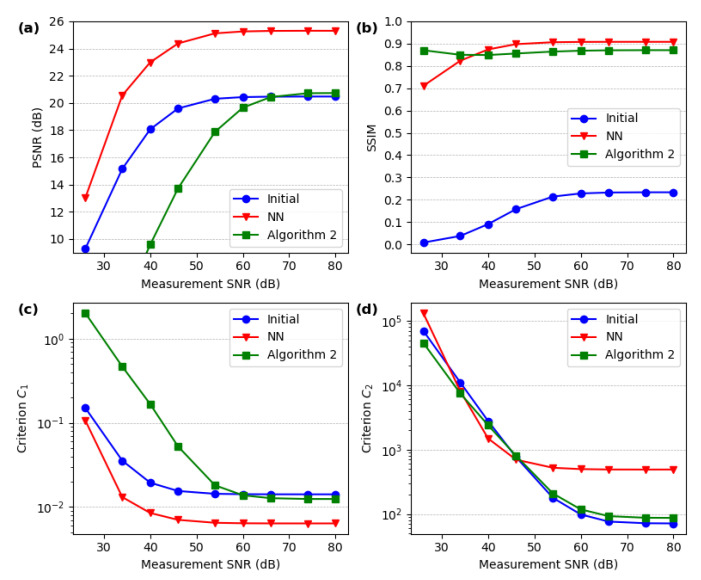
Sensitivity of the proposed image reconstruction methods to additive SPI measurement noise. We evaluate the quality of the initial reconstructions and the second-stage reconstructions, obtained with either Algorithm 2 or the NN, based on four criteria: (**a**) the peak signal-to-noise ratio (PSNR), (**b**) the structural similarity index (SSIM), and (**c**,**d**) the optimization criteria $C_{1}$ and $C_{2}$ from Equation (Equation 6). The metrics are averaged over a subset of 1000 images from the evaluation dataset, which consists of an equal mix of line art and handwritten digit images.

**Figure 13 sensors-24-08139-f013:**
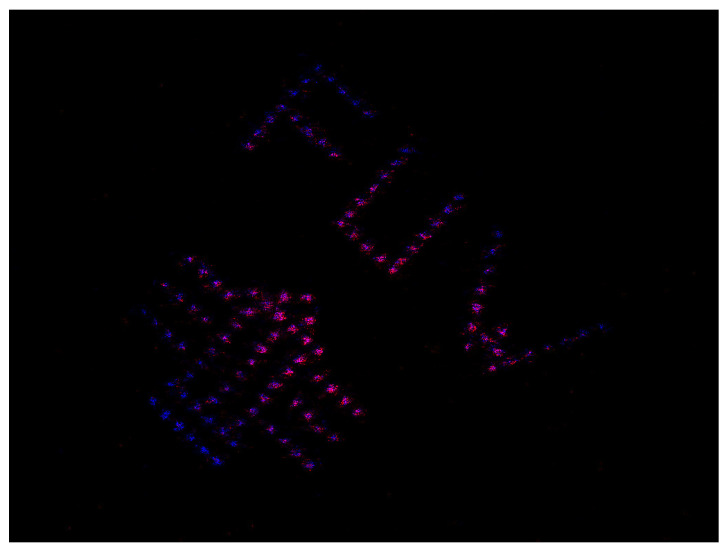
Experimental high-resolution SPI imaging. The object consists of a metal foil with holes arranged in the Faculty of Physics logo of the Univ. of Warsaw. The false colors, red and blue, correspond to the NIR and VIS channels, respectively. The object is static.

**Figure 14 sensors-24-08139-f014:**
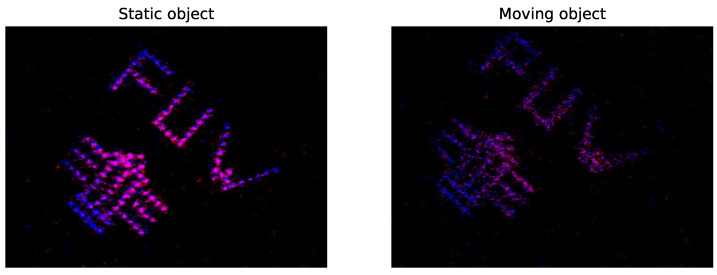
Experimental high-resolution SPI imaging of a static object (**left**) and of a moving object (**right**). For better visibility, the contrast of the false color images is enhanced. Object movement during the acquisition of the measurement vector results in deteriorated reconstruction quality and artifacts.

## Data Availability

Datasets with image maps, sampling patterns, generalized inverse matrix, NN model, etc., may be downloaded from the repository of the University of Warsaw [46]. We will release the Python code in a short time after the paper is published. The link to the code will be put in the repository [46]. Prior to this, we may provide parts of the code upon reasonable request. The code for generating image maps is similar to the one included in Supplement 1 of Ref. [32].

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
