# Peer review of "High-Resolution Single-Pixel Imaging of Spatially Sparse Objects: Real-Time Imaging in the Near-Infrared and Visible Wavelength Ranges Enhanced with Iterative Processing or Deep Learning"

_sensors, 2024, doi:10.3390/s24248139_

Round 1
Reviewer 1 Report
Comments and Suggestions for Authors
In the manuscript “High-resolution single-pixel imaging of spatially sparse objects: real-time imaging in the near-infrared and visible wavelength ranges enhanced with iterative processing or deep learning”, the authors used the full size of a 1024x768 DMD to modulate a same size object in VIS and IR bands, then reconstruct the object from 3200 SPI measurements. The object is spatially sparse. The authors also designed a set of binary patterns and proposed an iterative and a Unet reconstruction algorithms.
The reviewer has the following comments,
1. The descriptions about the motivation of the designed binary patterns and the iterative reconstruction method are not clear. Do the new designed patterns work better than random binary patterns? How did the authors get the image maps as shown in Fig.3? If the maps are object specific, then which kind of objects are for these maps? If the objects are changed, then should other kind of image maps be used? How to generate the new maps? The process to get a set of binary patterns from the image maps is complicated. The description is confusing.
2. Similarly, what is the advantage to use the regularized pseudoinverse as defined in Equation 2?
3. What is the compression noise? The noise introduced by a DAQ usually is the quantization error.
4. When 3200 SPI measurements are used for reconstructing a 1024x768 object, the compression ratio is 0.4069%. Why did the authors claim it as 0.34%?
Reviewer 2 Report
Comments and Suggestions for Authors
The manuscript is dedicated to a high-resolution, real-time single-pixel imaging method for spatially sparse objects. In this work authors have improved their algorithm, based on special MD-FDRI measurement matrices, and added an iterative reconstruction algorithm that enhances image quality while maintaining high processing speed for a extremely low sampling rate. I have a few suggestions that, in my opinion, will strengthen the manuscript and improve its clarity:
1. Firstly, I would recommend providing a clear and strict definition of what you consider "sparse" objects. Despite the presented image examples, the lack of a concrete definition will make it difficult to compare your results with those of other studies in the future. It would be useful to specify the upper limit at which your algorithm operates under the hardware resource constraints and real-time mode you have declared, and the lower bound that would make sense. For example, if a sparse object is considered to have only a single pixel in the image, it is easy to propose a much simpler and faster approach, such as Binary Space Partitioning, than your algorithm.
2. Additionally, I find it necessary to refer to related work [1], where the authors also obtain high-resolution images using DMD and discuss the possibility of reducing the required number of measurements (patterns) for sparse objects, but do not test it in practice.
[1] Li J. et al. Full-resolution, full-field-of-view, and high-quality fast Fourier single-pixel imaging // Opt. Lett. 2023. Vol. 48, No. 1. P. 49. doi.org/10.1364/OL.475956
Reviewer 3 Report
Comments and Suggestions for Authors
The manuscript titled "High-resolution single-pixel imaging of spatially sparse objects: real-time imaging in the near-infrared and visible wavelength ranges enhanced with iterative processing or deep learning" by Rafał Stojek et al., introduces a novel framework for high-resolution single-pixel imaging (SPI) that leverages a dedicated sampling scheme and an optimized reconstruction algorithm for rapid imaging of highly sparse scenes at the native digital micromirror device (DMD) resolution of 1024x768. This approach is technologically innovative, especially in its utilization of DMD for imaging at native resolutions. The manuscript is well-structured, logically progressing from introduction to materials and methods, followed by results and discussion, which is easy to follow. The figures and algorithm descriptions are clear and aid in the better understanding of the research content.
The paper meets the publication standards of the Sensors journal and is recommended for acceptance. The authors are encouraged to further highlight the advantages of their work over existing technologies and potential application scenarios in the final version to enhance the impact of the paper. The authors should ensure the accessibility of all data and code to allow reproducibility by other researchers, which would increase the transparency and credibility of the study. The manuscript presents a novel high-resolution single-pixel imaging technique, validates its effectiveness through experiments, and demonstrates its potential application across multiple wavelength ranges. The paper is clear, technically profound, and meets the publication criteria of the Sensors journal. Therefore, it is recommended for acceptance.
